# Low CD8^+^ T Cell Infiltration and High PD-L1 Expression Are Associated with Level of CD44^+^/CD133^+^ Cancer Stem Cells and Predict an Unfavorable Prognosis in Pancreatic Cancer

**DOI:** 10.3390/cancers11040541

**Published:** 2019-04-15

**Authors:** Ya-Chin Hou, Ying-Jui Chao, Min-Hua Hsieh, Hui-Ling Tung, Hao-Chen Wang, Yan-Shen Shan

**Affiliations:** 1Institute of Clinical Medicine, College of Medicine, National Cheng Kung University, Tainan 704, Taiwan; yachi2016@yahoo.com.tw (Y.-C.H.); surgeon.chao@gmail.com (Y.-J.C.); michelle1007825@hotmail.com (M.-H.H.); esr2wang@gmail.com (H.-C.W.); 2Department of Clinical Medical Research, National Cheng Kung University Hospital, College of Medicine, National Cheng Kung University, Tainan 704, Taiwan; 3Division of General Surgery, Department of Surgery, National Cheng Kung University Hospital, College of Medicine, National Cheng Kung University, Tainan 704, Taiwan; ritaiap.tw@gmail.com

**Keywords:** pancreatic cancer, T cells, cancer stem cells, CD8, PD-L1, CD44, CD133, immunotherapy

## Abstract

Cancer immunotherapy targeting immune checkpoints has exhibited promising clinical outcomes in many cancers, but it offers only limited benefits for pancreatic cancer (PC). Cancer stem cells (CSCs), a minor subpopulation of cancer cells, play important roles in tumor initiation, progression, and drug resistance. Accumulating evidence suggests that CSCs employ immunosuppressive effects to evade immune system recognition. However, the clinical implications of the associations among CD8^+^ T cells infiltration, programmed death receptor ligand-1 (PD-L1) expression, and CSCs existence are poorly understood in PC. Immunostaining and quantitative analysis were performed to assess CD8^+^ T cells infiltration, PD-L1 expression, and their relationship with CD44^+^/CD133^+^ CSCs and disease progression in PC. CD8^+^ T cells infiltration was associated with better survival while PD-L1 expression was correlated with PC recurrence. Both the low CD8^+^ T cells infiltration/high PD-L1 expression group and the high CD8^+^ T cells infiltration/high PD-L1 expression group show high levels of CD44^+^/CD133^+^ CSCs, but patients with low CD8^+^ T cells infiltration/high PD-L1 expression had worse survival and higher recurrence risk than those with high CD8^+^ T cells infiltration/high PD-L1 expression. Moreover, high infiltration of CD8^+^ T cells could reduce unfavorable prognostic effect of high co-expression of PD-L1 and CD44/CD133. Our study highlights an interaction among CD8^+^ T cells infiltration, PD-L1 expression, and CD44^+^/CD133^+^ CSCs existence, which contributes to PC progression and immune evasion.

## 1. Introduction

Pancreatic cancer (PC) is one of the most deadly cancers and the overall survival has improved only minimally over the past 40 years [1]. In 2018, PC was the seventh leading cause of cancer death in both males and females worldwide [2], and it is projected to become the second and third leading causes of cancer-related death in the United States as early as 2020 [3] and in the 28 countries of the European Union by the year 2025 [4], respectively. In Taiwan, according to the Ministry of Health and Welfare, PC has moved from the tenth leading cause of cancer death in 2006 to the eighth in 2016. Surgical resection offers the only realistic chance of cure for PC; unfortunately, less than 20% of patients present with resectable disease at the time of diagnosis [5]. Most treatment failures are due to local recurrence, local and/or distant micrometastases, or both and occur within two years after surgery [6,7]. Despite the fact that systemic chemotherapy is the mainstay of treatment in patients with unresectable PC, the treatment and survival outcomes remain poor [8,9]. Therefore, developing novel therapeutic strategies for this fatal disease is urgently needed.

Research in the field of cancer immunotherapy has been accelerating in recent years, due in part to immense clinical success of immune checkpoint blockade therapy for various types of cancer such as melanoma [10], Hodgkin’s lymphoma [11], head and neck squamous cell carcinoma [12], non-small cell lung cancer (NSCLC) [13], renal cell carcinoma [14], urothelial bladder carcinoma [15], and hepatocellular cancer [16]. These clinical trials are either already approved or are expected to be approved soon by the US Food and Drug Administration [17]. This therapeutic approach utilizes the cytotoxic T-lymphocyte-associated antigen 4 (CTLA-4) or programmed death 1 (PD-1)/programmed death receptor ligand-1 (PD-L1) blocking antibodies to release the brakes from suppressed T cells, allowing them to be activated and recover their antitumor activity [18], suggesting re-activation of antitumor immunity as a promising strategy for cancer treatment. Although it has been emerging as a new therapeutic approach beyond surgery, conventional chemotherapy, and radiation treatment, there has been limited success in the use of this approach for PC treatment which has been shown to be unresponsive to anti-CTLA-4 [19] and anti-PD-L1 [20].

The presence of CD8^+^ T cells within the tumor microenvironment (TME) or the invasive margin of the tumor, as well as the expression of PD-L1, have emerged to be the most powerful predictors for clinical outcome in response to checkpoint inhibition [21,22]. In PC, CD8^+^ T cells have been found in proximity to cancer cells and positively correlates with increased overall patient survival [23,24]. PD-L1 is the main ligand for PD-1, which is crucial for suppressing T cell proliferation, migration, and secretion of cytotoxic mediators, and thus blocks the antitumor immune responses by binding to PD-1 [25]. Several studies have evaluated the expression of PD-L1 on the surface of tumor cells, and suggested that PD-L1 expression can either relate to poor prognosis, better prognosis or show no association with prognosis for various malignancies including PC [8,26,27]. An early preclinical study has demonstrated that PD-L1 blockade promoted CD8^+^ T cell infiltration into the tumor and induced a substantial antitumor effect in a mouse tumor transplant model [28]. However, these preclinical findings have not translated to clinical success because no PC patients showed a clinical response in a phase I trial of anti-PD-L1 therapy [20]. This suggests that there may be other mechanisms contributing to the pro-tumorigenic immune phenotypes in PC.

Cancer stem cells (CSCs), a minor subpopulation of tumor cells, play important roles in tumor initiation, progression, and therapeutic resistance [29,30]. The existence of a CSC population establishes a functional hierarchy within a tumor tissue and encompasses both the self-renewal and differentiation hallmarks of stem cells [29]. Pancreatic CSCs have been identified based on a number of cell surface markers including CD24, CD44, CD133, ESA, c-Met, CXCR4, and ALDH1 [30]. With regard to their enhanced ability to establish and relapse of cancers, especially after chemotherapy and radiotherapy, CSCs have been suggested to have an advantage in evading immune detection and elimination [31]. The immune-suppressive mechanisms derived from CSCs interactions with the immune system are critical for CSCs to promote the expansion of pro-tumorigenic immune cells, which in turn regulates CSC maintenance and differentiation and thus sustains tumor establishment, growth, and metastasis [31,32]. Moreover, accumulating evidence indicates that T-cell activation costimulatory molecules are expressed at low levels, whereas T-cell inhibitory molecules including PD-L1 are expressed at high levels on CSCs [33,34]. These studies highlight the urgent need to understand the interplay between CSCs and the immune system. However, this interaction is complex and is only partially deciphered in PC.

Considering the importance of both CD8^+^ T cell infiltration and PD-L1 expression in defining the TME, we aimed to evaluate the feasibility of using this stratification as predictive biomarkers for clinical outcome of PC patients. To understand the regulation of cancer immune evasion and help further studies to develop improved immune-based therapeutic strategies, we also explored the relationship between this stratification and CD44^+^/CD133^+^ CSCs existence and determined their prognostic significance.

## 2. Results

### 2.1. CD8^+^ T Cells Infiltrate PC and Correlate with Patients’ Survival

We first performed IHC staining to validate the antibody used for detecting CD8^+^ T cells in PC specimens, and then evaluated the patterns of CD8^+^ T cell infiltration in TMA sections comprising 86 PC samples using a quantitative IF approach. Representative images displaying low or high CD8^+^ T cell infiltration are illustrated in Figure 1A,B. The density of CD8^+^ T cells was higher in the area around the tumor but lower at the center of the tumor. Kaplan-Meier survival analyses revealed that lower levels of CD8^+^ T cell infiltration were significantly associated with shorter OS (*p* = 0.0254; Figure 1C), but not with DFS (*p* = 0.3660; Figure 1D) in PC patients.

### 2.2. High PD-L1 Expression Is Associated with Recurrent PC

Considering that PD-L1 expression as a prognostic factor for the poor outcomes in many human cancers remains controversial, we sought to examine PD-L1 expression in our data set. IHC staining was performed to verify the primary antibody for detecting PD-L1 proteins. The PD-L1 expression levels were further measured by using a quantitative IF approach. Different expression levels of PD-L1 are illustrated in Figure 2A. The majority of PD-L1 was localized in tumor cells. High expression of PD-L1 was correlated with tumor recurrence (*p* = 0.0341; Figure 2B), but not with OS (*p* = 0.8899; Figure 2C) and DFS (*p* = 0.3889; Figure 2D).

### 2.3. Low CD8^+^ T Cell Infiltration and High PD-L1 Expression Predict an Unfavorable Prognosis in PC

As CD8^+^ T cells play a central role in adaptive anti-tumor immune responses and immune evasion associated with PD-L1 expression in tumors [35], we assessed CD8^+^ T cell infiltration, PD-L1 expression, and their relationship in PC. A significant positive correlation was observed between CD8^+^ T cells and PD-L1 expression (*r* = 0.5336, *p* < 0.0001; Figure 3A). Moreover, classification of tumors based on CD8^+^ T cells infiltration and PD-L1 expression has been proposed [21] and this concept has been applied to predict prognosis in patients with medulloblastoma [36], lung [37], and gallbladder cancer [38] by using immunostaining. We therefore performed IHC staining to study whether this phenomenon can be found in PC. Indeed, tumors could be categorized into 4 different types of tumors including low CD8^+^ T cells infiltration/low PD-L1 expression, high CD8^+^ T cells infiltration/low PD-L1 expression, low CD8^+^ T cells infiltration/high PD-L1 expression, and high CD8^+^ T cells infiltration/high PD-L1 expression. We also found that immunoreactivity for PD-1 was only detected in the area around the tumor as well as the area infiltrated by CD8^+^ T cells of the high PD-L1 expression group (Figure 3B). Using a quantitative IF approach, we classified the patients into 4 groups and found that patients with tumors exhibiting low CD8^+^ T cells infiltration and high PD-L1 expression had the worst OS (*p* = 0.0102; Figure 3C) and DFS (*p* = 0.0247; Figure 3D). Additionally, low CD8^+^ T cells infiltration (low group versus high group; hazard ratio (HR) = 0.511, 95% confidence interval (CI) = 0.281–0.930, *p* = 0.028 for OS) and low CD8^+^ T cells infiltration/high PD-L1 expression (low/low group versus low/high group; HR = 2.516, 95% CI = 1.104–5.735, *p* = 0.028 for OS; HR = 3.037, 95% CI = 1.373–6.716, *p* = 0.006 for DFS) were associated with poor survival in the univariate analysis. Multivariate analysis showed that low CD8^+^ T cells infiltration was an independent predictor for OS (Table 1).

### 2.4. Low CD8^+^ T Cell Infiltration and High PD-L1 Expression Are Associated with CSCs and Recurrent PC

CSCs are defined by their ability to self-renew and differentiate into cancer cells [39] and thus contribute to their functional role in the establishment and recurrence of cancerous tumors [40]. Our previous work has shown that pancreatic CSCs identified by CD44^+^/CD133^+^ expression had the ability to form tumorspheres, along with chemotherapeutic drug resistance and tumor-formation ability [41]. Moreover, increasing number of studies have reported that enriched PD-L1 expression in CSCs promotes immune evasion in head and neck [34], lung [42], and breast cancer [43]. As we have found that PD-L1 expression was associated with patients’ recurrence status and CD8^+^ T cells infiltration in Figure 2B and Figure 3A, we sought to analyze the correlations among PD-L1 expression, CD8^+^ T cells infiltration, and CSCs. TMA sections were double stained with antibodies against CSC markers CD44 and CD133, which were validated by a previous work [44]. The proportion of CD44^+^/CD133^+^ CSCs per patient was assessed by quantitative analysis. Notably, the level of CD44^+^/CD133^+^ CSCs was positively correlated with PD-L1 expression (*r* = 0.3577, *p* = 0.0007; Figure 4A) but was not with CD8^+^ T cells (*r* = 0.0697, *p* = 0.5235; Figure 4B). Consistent with these findings, the level of CD44^+^/CD133^+^ CSCs was significantly higher in tumor tissues with low CD8^+^ T cells infiltration/high PD-L1 expression or high CD8^+^ T cells infiltration/high PD-L1 expression than in those with low CD8^+^ T cells infiltration/low PD-L1 expression (*p* = 0.0006; Figure 4C). Interestingly, the percentage of recurrent PC in the patient group exhibiting low CD8^+^ T cells infiltration/high PD-L1 expression was the highest (85%) than that in those with high CD8^+^ T cells infiltration/high PD-L1 expression (70%), those with low CD8^+^ T cells infiltration/low PD-L1 expression (60%), and those with high CD8^+^ T cells infiltration/low PD-L1 expression (61%; *p* = 0.0004; Figure 4D).

### 2.5. Unfavorable Prognostic Value of High Co-Expression of PD-L1 and CD44/CD133 Is Eliminated by High CD8^+^ T Cell Infiltration

As shown in Figure 2B–D and Figure 4A, PD-L1 expression was not associated with survival, but was positively associated with PC recurrence and CD44^+^/CD133^+^ CSCs. Our previous work has demonstrated that high CD44/CD133 expression was correlated with shorter survival in PC patients [44]. Therefore, we evaluated the rates of OS and DFS based on PD-L1 and CD44/CD133 expression. Kaplan-Meier survival analyses showed that patients with high co-expression of PD-L1 and CD44/CD133 had significantly worse OS (*p* = 0.0306; Figure 5A) and DFS (*p* = 0.0043; Figure 5B) than the other patients. Although a high level of CD44^+^/CD133^+^ CSCs was present in low CD8^+^ T cells infiltration/high PD-L1 expression group as well as in high CD8^+^ T cells infiltration/high PD-L1 expression group, patients with low CD8^+^ T cells infiltration and high PD-L1 expression had the worse survival rates and higher risk of recurrence than those with high CD8^+^ T cells infiltration and high PD-L1 expression as shown in Figure 3C,D and Figure 4C,D. We next investigated whether CD8^+^ T cells infiltration can affect the prognostic value of co-expression of PD-L1 and CD44/CD133 in PC patients. Among the 28 patients who had high co-expression of PD-L1 and CD44/CD133, those in the low CD8^+^ T cells infiltration group showed a significantly worse OS than those in the high CD8^+^ T cells infiltration group (*p* = 0.0281; Figure 5C). These data suggested that the unfavorable prognostic effect of high co-expression of PD-L1 and CD44/CD133 can be counteracted by high CD8^+^ T cells infiltration. In addition, we also observed that there was no difference in the DFS between high co-expression of PD-L1 and CD44/CD133 cases with low and high CD8^+^ T cells infiltration (*p* = 0.2774; Figure 5D). The results are consistent with those findings that CD8^+^ T cells infiltration only affect OS but not DFS of PC patients shown in Figure 1C,D.

## 3. Discussion

In this study, we showed that CD8^+^ T cells infiltration was associated with longer survival while PD-L1 expression was correlated with PC recurrence. When categorized into 4 different types of TME based on CD8^+^ T cells infiltration and PD-L1 expression, pancreatic tumors with low CD8^+^ T cells infiltration and high PD-L1 expression had the worst survival rates and the highest risk of recurrence. These worse outcomes were related to the existence of CD44^+^/CD133^+^ CSCs. High levels of PD-L1 expression and CD44^+^/CD133^+^ CSCs correlates with worse survival and its unfavorable prognostic effect could be reduced by high CD8^+^ T cell infiltration.

Translational applicability of classifying cancers based on CD8^+^ T cells infiltration and PD-L1 expression into the clinical setting of oncological diseases has been proposed. Among the 4 groups of gallbladder cancer, there were significant differences in both OS and progression-free survival (PFS), wherein the patients with PD-L1^−^ and CD8^high^ tumor- infiltrating lymphocytes (TILs) had the best survival, and the patients with PD-L1^+^ and CD8^low^ TILs had the worst survival [38]. NSCLC patients with low CD8 expression and high PD-L1 expression had very poor outcomes, with a median OS of 3.7 months, while the median for all other patients was 8 months [37]. In human medulloblastoma, patients with high expression of PD-L1 and low infiltration of CD8^+^ lymphocytes had a significantly worse outcome, with a 5-year survival rate of 15%, as compared with the other patients, who had a 5-year survival rate of nearly 90% [36]. Neither expression of PD-L1 nor density of CD8^+^ T cells was associated with OS or PFS; however, there was a trend towards worse PFS in cervical cancer patients whose tumors expressed PD-L1 but lacked CD8^+^ T cells [45]. Here we observed that the percentage of recurrent PC was the highest in the low CD8^+^ T cells infiltration/high PD-L1 expression group (85%, *n* = 13), followed by the high CD8^+^ T cells infiltration/high PD-L1 expression group (70%, *n* = 30), the high CD8^+^ T cells infiltration/low PD-L1 expression group (61%, *n* = 13), and the low CD8^+^ T cells infiltration/low PD-L1 expression group (60%, *n* = 30). Although there was little difference among these groups and the results may be influenced by the small sample sizes, we found that PC patients with low CD8^+^ T cells infiltration and high PD-L1 expression experienced the worst OS and DFS, which is consistent with a previous study using a PC database from The Cancer Genome Atlas [46]. Indeed, the combination of CD8^+^ T cells infiltration and high PD-L1 expression stratifies patient survival.

Several studies focused on PC have achieved similar results showing that PD-L1 expression in tumor tissues is associated with tumor growth, TNM stage, and patients’ survival [8,26]. In our study, although PD-L1 expression was not correlated with patients’ survival, it was associated with PC recurrence. Collectively, PD-L1 expression in tumor tissues is associated with poor clinical outcomes. We also observed that PD-L1 expression was positively correlated with CD8^+^ T cells infiltration. Similar results were also reported in medulloblastoma, hepatocellular carcinoma, breast, and prostate cancer [36,47,48,49]. These paradoxical results were found between prognostic and correlation analysis of CD8^+^ T cells infiltration and PD-L1 expression, suggesting that CD8^+^ T cells become more exhausted and dysfunctional during tumorigenesis of these cancer types. Therefore, our data showed that CD8^+^ T cells infiltration only affect OS but not DFS of PC patients, which might be one of the causes for the limited success in the use of immune checkpoint inhibitors for PC treatment.

Critical obstacles to immunotherapy in PC can be explained by the low immunogenicity and immunosuppressive TME of these tumors [8,50]. The TME mainly consists of stromal cells including fibroblasts, immune and inflammatory cells, the blood and lymphatic vascular networks, and extracellular matrix [51]. Among the immune and inflammatory cells, CD8^+^ T cells and CD4 helper T cells 1 (TH1), NK cells, M1 macrophages, dendritic cells work against the tumor while regulatory T cells, M2 macrophages, myeloid derived suppressor cells (MDSC), CD4 helper T cells 2 (TH2) cells promote tumor growth [52]. Dynamic interplay between cancer cells and immune cells can provoke pro-tumorigenic behavior and thus drive tumor growth, invasion, metastasis, and immune suppression [51,53]. In PC, tumor-educated macrophages directly enhance the level of CSCs, thereby facilitating macrophage-mediated suppression of CD8^+^ T cells infiltration [54]. Pancreatic TME can induce monocyte transformation to MDSC expansion, which increases in epithelial-mesenchymal transition of tumor cells and the level of CSCs [55]. Of note, CSCs not only promote the expansion of pro-tumorigenic immune cells [31] but also actively suppress T cells activation [56,57]. Our preliminary data found that PD-L1 expression was higher in CSCs that expressed high levels of the CSCs marker CD44, CD133, ALDH, and ABCG2 than in non-CSCs isolated from PANC-1 and MIA PaCa-2 PC cells. We also performed the co-culture experiments of CSCs and CD8^+^ T cells sorted from PBMCs treated with PHA (10 μg/mL; Sigma, St. Louis, MO, USA) to evaluate the ability of CSCs to escape from immune surveillance by using cell viability assay. In non-CSCs group, co-culture with CD8^+^ T cells led to a significant reduction in viability, whereas there was no significant difference in the percentages of cell viability between CSCs and CSCs co-cultured with CD8^+^ T cells (data not shown), suggesting that CSCs may possess enhanced ability to evade immune detection and elimination in comparison with non-CSCs. As expected, we found that the levels of CD44^+^/CD133^+^ CSCs were correlated with PD-L1 expression, and their combination conferred unfavorable prognosis.

Cancer immunotherapy targeting immune checkpoints has exhibited promising clinical outcomes in various types of solid tumors, but it offers only limited benefits for PC. The efficacy of immune checkpoint inhibitors depends on the presence of an endogenous antitumor T cells response. This requires not only the generation of cancer-specific T cells but also the physical contact of T cells with cancer cells [58]. Blocking the PD-1/PD-L1 pathway has been demonstrated to reactivate and increase infiltration of T cells into some cancers [50,59]. As multiple studies have indicated that PD-L1 expression was increased in human pancreatic tumors [8], suggesting a certain efficacy of these inhibitors in the treatment of PC. Most importantly, compared with other malignancies, the extensive hypovascularity of stroma surrounding cancer cells is a prominent pathological characteristic of PC [60], which may favor CSCs and the CSC-mediated expansion of pro-tumorigenic immune cells to resist to these inhibitors. For this reason, our study also focused on determining the level of CD44^+^/CD133^+^ CSCs in different types of TME based on CD8^+^ T cells infiltration and PD-L1 expression. Our study highlights an interaction among CD8^+^ T cells infiltration, PD-L1 expression, and CD44^+^/CD133^+^ CSCs existence, thus providing a rationale for designing ideal combination cancer therapies based on tumor immunology.

## 4. Materials and Methods

### 4.1. Patients and Samples

Tissue specimens collected from patients with PC who underwent surgical resection was approved by the Institutional Review Board of National Cheng Kung University Hospital (Tainan, Taiwan). The number of IRB approval is A-ER-105-459. Informed consent was obtained from all patients. Clinical information from all PC patients between 2001 and 2012 were reviewed using electronic medical records (Table 2). The PC tissue microarrays (TMAs) were constructed from formalin-fixed paraffin- embedded (FFPE) blocks of 86 archived PC specimens using the method described previously [61]. Briefly, representative areas of tumor tissues were selected based on the review of hematoxylin and eosin-stained slides by an experienced pathologist. The corresponding FFPE tissue blocks were retrieved. For each patient, two or four 2 mm cores from representative areas of the tumors were used for TMA construction.

### 4.2. Hematoxylin and Eosin (H&E), Immunohistochemistry (IHC), and Immunofluorescence (IF) Staining and Quantitative Analysis

FFPE PC tissues or TMA blocks were cut into 5-μm-thick sections and stained by H&E or stained with anti-human CD8 antibody (1:100 dilution; M7103, DAKO, Carpinteria, CA, USA), anti-human PD-L1 antibody (1:50 dilution; ab205921, Abcam, Cambridge, MA, USA), anti-human PD-1 antibody (1:50 dilution; ab117420, Abcam), anti-human CD44 antibody (1:500 dilution; M7082, DAKO), or anti-human CD133 antibody (1:500 dilution; 3663S, Cell Signaling, Danvers, MA, USA) at 4 °C overnight. TMA sections were followed by incubation with appropriate fluorescently conjugated secondary antibodies for IF at room temperature for 1 hour and 4′,6-diamidino-2-phenylindole (DAPI) was used to stain the nucleus. Slides for immunoreaction products of IHC were followed by incubation HRP-conjugated secondary antibodies, developed with DAB chromogen system (DAKO), and counterstained with hematoxylin. The full-field images of IHC-stained slides or IF-stained TMAs were acquired using the FACS-like Tissue Cytometry system (Tissue Gnostics, Vienna, Austria). The percentage of CD8-, PD-L1-, or CD44/CD133-positive cells in each fluorescence image was quantified and calculated the average of cell percentages in duplicate or quadruplicate cores per patient by TissueQuest system and the software TissueFAX (Tissue Gnostics, Vienna, Austria) as described previously [61].

### 4.3. Statistical Analysis

Statistical analyses were performed using the GraphPad Prism 5.0 (GraphPad Software Inc., La Jolla, CA, USA) and SPSS Statistics 17.0 (IBM, Endicott, NY USA) software. The median value of CD8, PD-L1, or CD44/CD133 expression in the tumor samples was used to stratify PC patients into the low-expression group (≤median) or the high-expression group (>median). The survival probability was calculated with the Kaplan-Meier method. Univariate and multivariate analyses with a Cox proportional hazards model or logistic regression model were performed to assess significant factors. Data were expressed as the means ± standard error in all experiments. *p* values less than 0.05 were considered significant.

## 5. Conclusions

The combination of CD8^+^ T cells infiltration and PD-L1 expression is well-suited as an indicator for classifying PC tumors and predicting clinical outcomes. Low CD8^+^ T cell infiltration and high PD-L1 expression are correlated with the level of CD44^+^/CD133^+^ CSCs. Our study highlights an interaction among CD8^+^ T cells infiltration, PD-L1 expression, and CD44^+^/CD133^+^ CSCs existence, which contributes to PC progression and immune evasion. These findings offer the potential factors for prognostic evaluation of PC and help in designing promising immune-based therapeutic strategies.

## Figures and Tables

**Figure 1 cancers-11-00541-f001:**
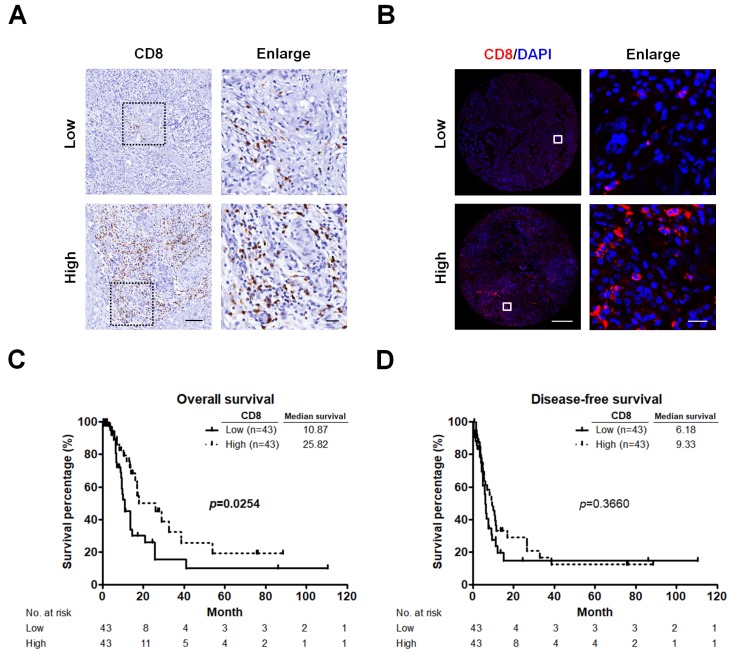
CD8^+^ T cells significantly correlate with survival. (**A**) Representative pictures for low and high CD8 immunostaining (brown) in PC tumors. Original magnification, 20×; scale bars, 100 μm. Enlarged images on the right panel are shown for the areas outlined by black squares. Scale bars show 25 μm in length. (**B**) Representative images showing TMA cores from two different PC tumors after IF analysis of CD8 (red) and nuclear (4′,6-diamidino-2-phenylindole, DAPI; bule) staining. Original magnification, 20×; scale bars, 400 μm. Enlarged images on the right panel are shown for the areas outlined by white squares. Scale bars show 20 μm in length. (**C**,**D**) Kaplan-Meier survival curves showing comparison of OS (**C**) and DFS (**D**) between low and high infiltration of CD8^+^ T cells. *p* values determined using the log-rank test. OS, overall survival; DFS, disease-free survival.

**Figure 2 cancers-11-00541-f002:**
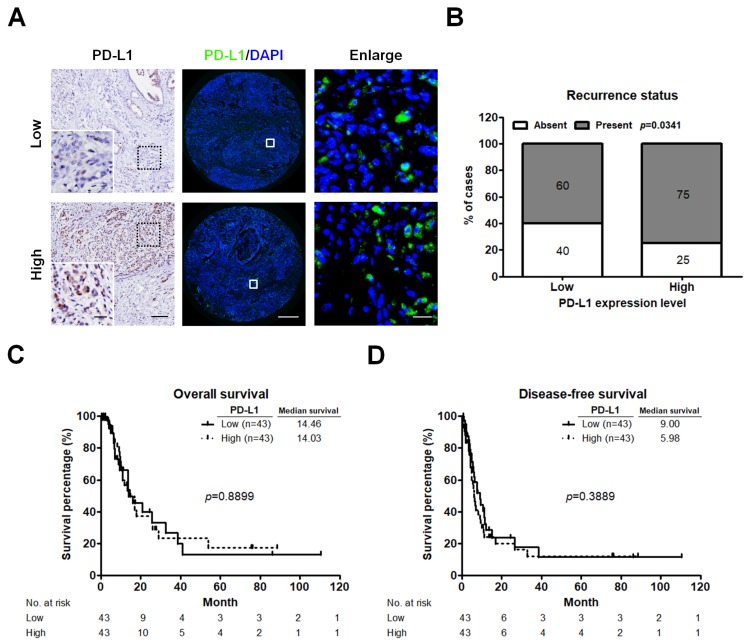
PD-L1 expression associates with patients’ recurrence status. (**A**) IHC and IF images displaying low and high level of PD-L1 staining in PC tumors. IHC staining (left panel) of PD-L1 is shown as brown colored sections. Original magnification, 20×; scale bars, 100 μm. Enlarged images on the left lower panel are shown for the areas outlined by black squares. Scale bars show 25 μm in length. IF staining (middle panel) of PD-L1 in TMA cores from two different PC tumors. Green indicates PD-L1 staining. The nuclei were stained by 4′,6-diamidino-2-phenylindole (DAPI, bule). Enlarged images on the right panel are shown for the areas outlined by white squares in middle panel. Scale bars show 20 μm in length. (**B**) Relative distribution analysis of PC patients’ recurrence status within low (*n* = 43) or high PD-L1 expression group (*n* = 43). Absent, patients without recurrent PC; present, patients with recurrent PC. *p* values calculated using Chi-square test. (**C**,**D**) Survival analysis of PC patients stratified by PD-L1 expression level using the Kaplan-Meier estimator to determine the OS (**C**) and DFS (**D**). *p* values determined using the log-rank test. OS, overall survival; DFS, disease-free survival.

**Figure 3 cancers-11-00541-f003:**
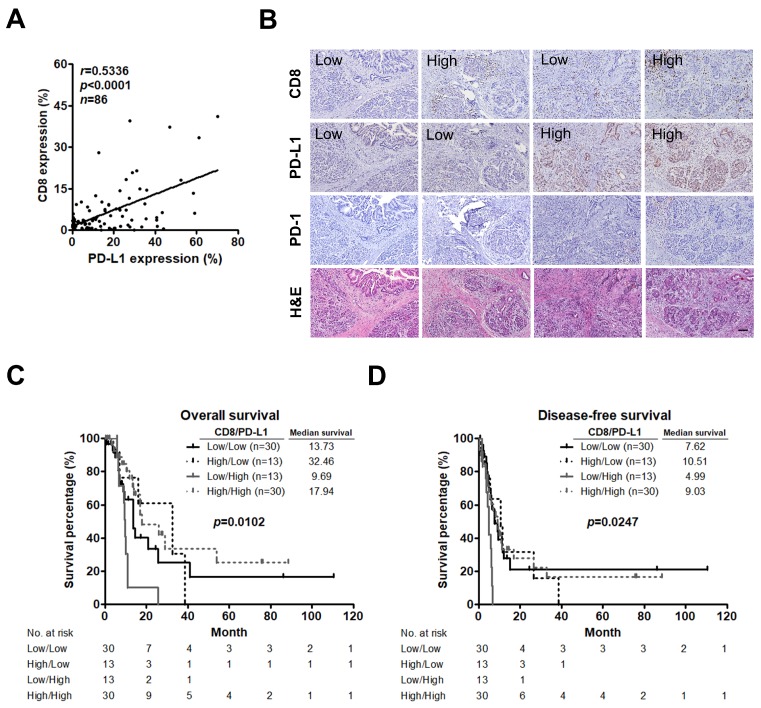
Low CD8^+^ T cell infiltration and high PD-L1 expression significantly stratifies patient survival. (**A**) Correlation analysis between percent CD8^+^ T cells and PD-L1 expression per patient. Pearson correlation coefficient (r) and significance level (*p* value) is shown for correlation. (**B**) IHC staining of CD8^+^ T cells infiltration (*upper panel*), PD-L1 expression (*second panel*), PD-1 expression (*third*
*panel*), and H&E staining (*lower panel*) in the same tumor tissues. Original magnification, 20×; scale bars, 100 μm. (**C**,**D**) Kaplan-Meier curves for OS (**C**) and DFS (**D**) of patients stratified based on CD8^+^ T cells infiltration and PD-L1 expression. *P* values determined using the log-rank test. OS, overall survival; DFS, disease-free survival.

**Figure 4 cancers-11-00541-f004:**
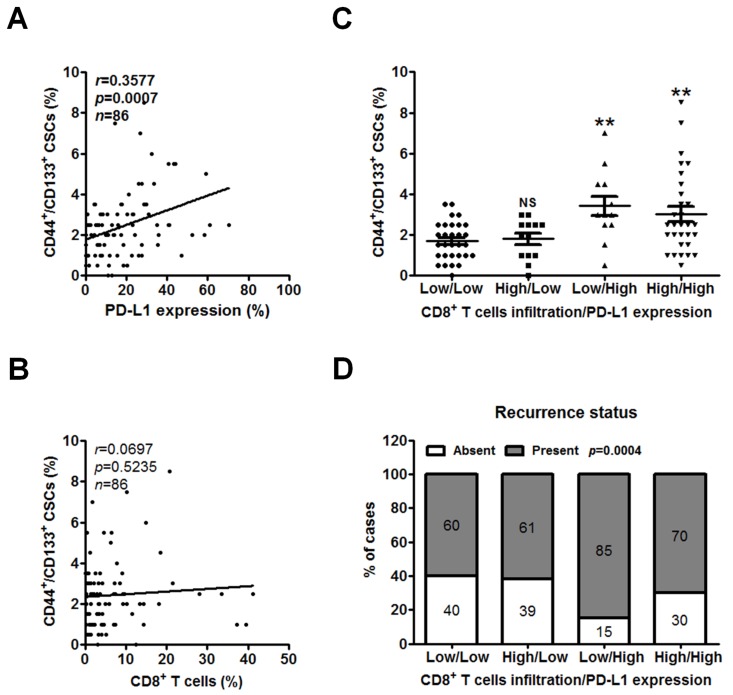
Low CD8^+^ T cell infiltration and high PD-L1 expression are associated with CSCs and related to the recurrence of PC. (**A**,**B**) The PC TMAs were double IF stained for CSC markers CD44 and CD133 and further assessed by quantitative analysis. Correlation of percent CD44^+^/CD133^+^ CSCs with PD-L1 expression (**A**) or percent CD8^+^ T cells (**B**) per patient was analyzed by Pearson’s correlation. (**C**,**D**) Classification of tumors based on CD8^+^ T cells infiltration and PD-L1 expression into 4 groups including low CD8^+^ T cells infiltration/low PD-L1 expression (*n* = 30), high CD8^+^ T cells infiltration/low PD-L1 expression (*n* = 13), low CD8^+^ T cells infiltration/high PD-L1 expression (*n* = 13), and high CD8^+^ T cells infiltration/high PD-L1 expression (*n* = 30). (**C**) Dot plots showing the percentage of CD44^+^/CD133^+^ CSCs in each group. All values are the mean ± s.e.m. NS, not significant; **, *p* < 0.01 compared with low CD8^+^ T cells infiltration/low PD-L1 expression group, as determined using one-way ANOVA. (**D**) Relative distribution analysis of PC patients’ recurrence status within each group. Absent, patients without recurrent PC; present, patients with recurrent PC. *p* values calculated using Chi-square test. CSCs, cancer stem cells.

**Figure 5 cancers-11-00541-f005:**
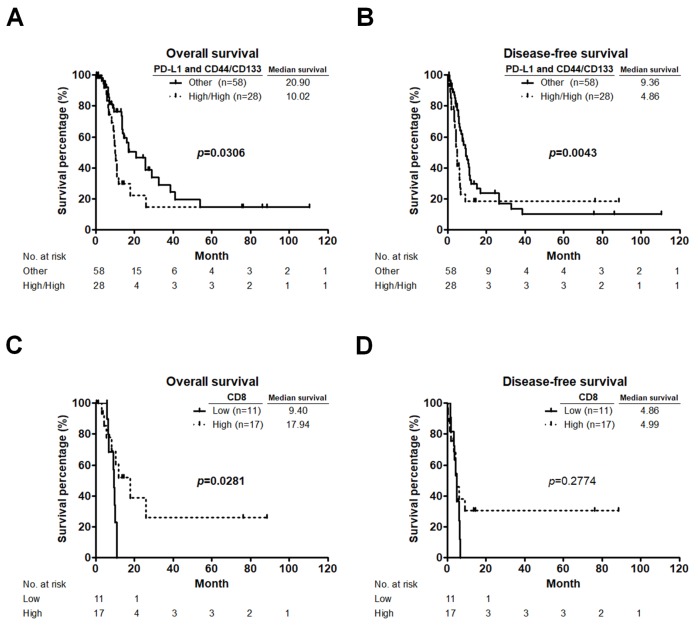
CD8^+^ T cells infiltration affects the prognostic value of co-expression of PD-L1 and CD44/CD133 in PC patients. (**A**,**B**) OS (**A**) and DFS (**B**) curves for patients with high co-expression of PD-L1 and CD44/CD133 compared to those with either low co-expression of PD-L1 and CD44/CD133, low PD-L1 expression and high CD44/CD133 expression, or high PD-L1 expression and low CD44/CD133 expression (other group). *p* values determined using the Gehan-Breslow-Wilcoxon test. (**C**,**D**) OS (**C**) and DFS (**D**) curves for patients with low and high CD8^+^ T cells infiltration based on high co-expression of PD-L1 and CD44/CD133. *p* values determined using the log-rank test. OS, overall survival; DFS, disease-free survival.

**Table 1 cancers-11-00541-t001:** Univariate and multivariate analysis of CD8^+^ T cell infiltration and PD-L1 expression for overall and disease-free survival.

Variable	*n*	Univariate				Multivariate			
		OS		DFS		OS		DFS	
		HR (95% CI)	*p*	HR (95% CI)	*p*	HR (95% CI)	*p*	HR (95% CI)	*p*
**CD8^+^ T cell infiltration**									
Low	43	Reference		Reference		Reference		Reference	
High	43	0.511 (0.281–0.930)	**0.028**	0.787 (0.467–1.327)	0.369	0.378 (0.182–0.788)	**0.009**	0.571 (0.298–1.093)	0.091
**PD-L1 expression**									
Low	43	Reference		Reference		Reference		Reference	
High	43	0.959 (0.533–1.726)	0.890	1.254 (0.747–2.106)	0.391	1.665 (0.813–3.408)	0.163	1.274 (0.909–3.270)	0.095
**CD8^+^ T cell infiltration/PD-L1 expression**									
Low/Low	30	Reference		Reference					
High/Low	13	0.778 (0.283–2.139)	0.626	1.118 (0.485–2.578)	0.793				
Low/High	13	2.516 (1.104–5.735)	**0.028**	3.037 (1.373–6.716)	**0.006**				
High/High	30	0.615 (0.299–1.264)	0.186	0.994 (0.527–1.872)	0.984				

Values in boldface indicate *p* < 0.05. Abbreviations: *n*, number of patients; OS, overall survival; DFS, disease-free survival; HR, hazard ratio; CI, confidence interval.

**Table 2 cancers-11-00541-t002:** Clinical parameters of pancreatic cancer patients in TMAs.

Variable	No. of Patients (%)
**Sex**	
Men	56 (65.1)
Women	30 (34.9)
**Age**	
≤65	43 (50.0)
>65	43 (50.0)
**Tumor location**	
Head	55 (64.0)
Neck	6 (7.0)
Body/tail	13 (15.1)
Uncinate process	12 (14.0)
**Tumor size, cm**	
≤3	48 (55.8)
>3	38 (44.2)
**Margin status**	
R0	60 (69.8)
R1	22 (25.6)
R2	4 (4.7)
**Tumor grade**	
Poorly differentiated	16 (18.6)
Moderately differentiated	47 (54.7)
Well differentiated	23 (26.7)
**Stage**	
I	11 (12.8)
II	70 (81.4)
III	3 (3.5)
IV	2 (2.3)
**Adjuvant therapy**	
Yes	33 (38.4)
No	53 (61.6)
**CA19-9, U/mL**	
≤37	18 (20.9)
>37	68 (79.1)

Abbreviations: TMAs, tissue microarrays; CA19-9, carbohydrate antigen 19-9.

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
