# Peer review of "Low CD8+ T Cell Infiltration and High PD-L1 Expression Are Associated with Level of CD44+/CD133+ Cancer Stem Cells and Predict an Unfavorable Prognosis in Pancreatic Cancer"

_cancers, 2019, doi:10.3390/cancers11040541_

Round 1

Reviewer 1 Report

The article “low CD8+ T cell infiltration and high PD-L1 expression are associated with CD44+/CD133+ cancer stem cells and predict an unfavorable prognosis in pancreatic cancer” by Ya-Chin Hou et al., describes CD8+ T cells infiltration was associated with longer survival while PD-L1 expression was correlated with pancreatic cancer recurrence. Especially, they represented that High levels of PD-L1 expression and CD44+/CD133+CSCs correlates with worse survival and its unfavorable prognostic effect could be reduced by high CD8+ T cell infiltration. However, it is well-known that the progression effect of PD-L1 and anticancer effect by CD8+ T cell infiltration in several cancers. Moreover, it was previously reported that PD-L1 promotes Oct4 and nanog expression in breast cancer stem cells (IJC 2017,141, 1402-1412), and PD-L1 accumulation on cancer stem cells promotes immune evasion (Nat commun. 2018, 9, 1908). Therefore, this manuscript has a weak novelty. But, due to address the roles of PD-L1 and CD8+ T cell infiltration on pancreatic cancer stem cells on the basis of patient’ data, it is interest. However, we cannot convince whether the high PD-L1 expression result in immune evasion in pancreatic cancer stem cells. So, it need to experimental data. For example, to address the CD8+ T cell infiltration and role of PD-L1 in cancer stem cells, we simply can use the the co-culture experiment; non-cancer stem cell population and cancer stem cell population were co-cultured with CD8+ T cells, and confirm the cell viability and PD-L1 expression level in each population. Via the experiment, we can know that the low survival and recurrence in pancreatic cancer is induced by the reduction of CD8+ T cell infiltration, contributing to immune evasion resulted in highly expressed PD-L1 on cancer stem cells.

Author Response

Reviewer #1

The article “low CD8+ T cell infiltration and high PD-L1 expression are associated with CD44+/CD133+ cancer stem cells and predict an unfavorable prognosis in pancreatic cancer” by Ya-Chin Hou et al., describes CD8+ T cells infiltration was associated with longer survival while PD-L1 expression was correlated with pancreatic cancer recurrence. Especially, they represented that High levels of PD-L1 expression and CD44+/CD133+CSCs correlates with worse survival and its unfavorable prognostic effect could be reduced by high CD8+ T cell infiltration. However, it is well-known that the progression effect of PD-L1 and anticancer effect by CD8+ T cell infiltration in several cancers. Moreover, it was previously reported that PD-L1 promotes Oct4 and nanog expression in breast cancer stem cells (IJC 2017,141, 1402-1412), and PD-L1 accumulation on cancer stem cells promotes immune evasion (Nat commun. 2018, 9, 1908). Therefore, this manuscript has a weak novelty. But, due to address the roles of PD-L1 and CD8+ T cell infiltration on pancreatic cancer stem cells on the basis of patient’ data, it is interest. However, we cannot convince whether the high PD-L1 expression result in immune evasion in pancreatic cancer stem cells. So, it need to experimental data. For example, to address the CD8+ T cell infiltration and role of PD-L1 in cancer stem cells, we simply can use the the co-culture experiment; non-cancer stem cell population and cancer stem cell population were co-cultured with CD8+ T cells, and confirm the cell viability and PD-L1 expression level in each population. Via the experiment, we can know that the low survival and recurrence in pancreatic cancer is induced by the reduction of CD8+ T cell infiltration, contributing to immune evasion resulted in highly expressed PD-L1 on cancer stem cells.

Ans: Thank you for your comment and suggestion. We have performed the co-culture experiments of cancer stem cells (CSCs) and CD8+ T cells sorted from PBMCs treated with PHA (10 μg/ml; Sigma, St. Louis, MO) to evaluate the ability of CSCs to escape from immune surveillance by using cell viability assay. In non-CSCs group, co-culture with CD8+ T cells led to a significant reduction in viability, whereas there was no significant difference in the percentages of cell viability between CSCs and CSCs co-cultured with CD8+ T cells (see below). The description has been added in page 6, lines 242-247, but the figure was not shown in the manuscript. We also confirmed that PD-L1 expression was higher in CSCs than in non-CSCs isolated from PANC-1 and MIA PaCa-2 cells. The results suggest that high expression of PD-L1 allows CSCs to evade immune detection and elimination in comparison with non-CSCs.

(A) The bright field micrographs showed sphere formation of CD44+/CD133+ cancer stem cells (CSCs) isolated from PANC-1 or MIA PaCa-2 cells cultured in ultra-low attachment plates for 14 days. Original magnification, 100x, scale bars, 100 μm. (B) PD-L1 mRNA expression in non-CSCs and CSCs population of PANC-1 and MIA PaCa-2 cells was measured by qPCR and is depicted as fold changes relative to each non-CSCs. *, P<0.05; **, P<0.01 compared with non-CSCs, as determined using two-way ANOVA. (C) The viability of non-CSCs and CSCs population of PANC-1 (left) and MIA PaCa-2 (right) co-cultured with or without CD8+ T cells was analyzed using MTT assays after 24 hours of incubation. NS, not significant; *, P<0.05; ***, P<0.001 compared with monoculture (mono) group in non-CSCs and CSCs population, as determined using two-way ANOVA.

Reviewer 2 Report

Hou et al. showed the clinical implications of the associations among CD8+ T cells infiltration, PD-L1 and CSCs in PC via immunostaining etc.

This is interesting to get an insight of immune response, PD-L1 expression and CSCs.

However, it requires revision for the publication of “Cancers” as below.

PD-1 expression is also important to check via IF or immunostaining as well as PD-L1 in PC.

It requires clearer table 1 to see as like table 2.

In Materials and methods, IRB number should be included.

Miscellaneous, in page 13, line 512, 'figure legends' should be deleted.

Author Response

Reviewer #2

Hou et al. showed the clinical implications of the associations among CD8+ T cells infiltration, PD-L1 and CSCs in PC via immunostaining etc. This is interesting to get an insight of immune response, PD-L1 expression and CSCs. However, it requires revision for the publication of “Cancers” as below.

1)     PD-1 expression is also important to check via IF or immunostaining as well as PD-L1 in PC.

Ans: Thank you for your comment and suggestion. We have performed IHC staining to check PD-1 expression in 4 different types of tumors including low CD8+ T cells infiltration/low PD-L1 expression, high CD8+ T cells infiltration/low PD-L1 expression, low CD8+ T cells infiltration/high PD-L1 expression, and high CD8+ T cells infiltration/high PD-L1 expression. We found that immunoreactivity for PD-1 was only detected in the area around the tumor as well as the area infiltrated by CD8+ T cells of the high PD-L1 expression group. Please check page 3, lines 127-129, page 8, line 284, Figure 3B, and figure legends (page, 16, lines 557-560): Changed as recommended.

2)     It requires clearer table 1 to see as like table 2.

Ans: Thank you for your comment. Table 1 was revised as recommended. Please check it in page 4.

3)     In Materials and methods, IRB number should be included.

Ans: Thank you for your comment. The number of IRB approval, A-ER-105-459, has been added in the manuscript as recommended. Please check page 7, line 270.

4)     Miscellaneous, in page 13, line 512, 'figure legends' should be deleted.

Ans: Thank you for your suggestion. Page 13, line 529: The “figure legends” was deleted. Please check it.

Reviewer 3 Report

In this article Hou et al., report that low infiltration of CD8+ T cells in relation to high expression of PD-L1 are associated tumor recurrence in patients with pancreatic cancer (PC). Authors also show that low infiltration of T cells/high expression of PD-L1 is associated with levels of CD44+/CD133+ cancer stem cells and conclude that this correlation of infiltration/high expression in association with CD44+/CD133+ cancer stem cells could help predict an unfavorable prognosis in patients with PC.  

The current study has a scope in a view that the study is performed on patient samples and expression correlates with tumor recurrence in patients and survival rate. The experimental design is technically sound and conclusions are supported with experimental evidence. However, there are some points that authors need to revise. 

1.      The part of title which says ‘’….expression are associated with cancer stem cells and…’’. Indicate what capacity of cancer stem cells?. i.e. use an appropriate biological term with cancer stem cells e.g. any of these: cancer stem cell initiation, or cancer stem cell expression, or cancer stem cell levels, or cancer stem cell expansion.

2.      Same at page 4, line 135: CD44+/CD133+ CSCs, and Line 145; CD44+/CD133+ CSCs. Apply appropriate term with CSCs throughout the manuscript, including figure legends.

3.      Line 31 of abstract: ‘’Moreover, CD8+ T cells infiltration could reduce unfavorable prognostic effect of high co-expression of PD-L1 and CD44/CD133’’. Perhaps better to mention high infiltration (high infiltration of CD8+ T cells).

4.      Page 5, line 153; ‘’percentage of recurrent PC in patients with tumors exhibiting low CD8+ T cells infiltration….’’. Authors may wish to rephrase it as ‘’percentage of recurrent PC in patients group exhibiting low……’’.

5.      Figure 2 B. were the number of patients same in both low PD-L1 expression group and low PD-L1 expression group for evaluating the percentage of recurrence. In legends, mention the total number of patients included in each group.

6.      In Figure 4C (CD8 T cell infiltration/PD-L1 expression; low/high panel for CD 44/CD133 CSCs) and Figure 4D (CD8 T cell infiltration/PD-L1 expression; low/high panel for recurrence): Does this low/high panel in figure 4D correspond to same NUMBER of patients in 4C?. Perhaps mentioning the number of patients in 4D will be a suitable presentation, (however Figure 4C is self/explanatory by dot plots). This is important to make reader understandable whether this panel low/high in 4D corresponds to same patients and same NUMBER of patients when it comes to represent percentage.  

7.      Figure 4D high/high panel shows 70% of the cases exhibit recurrence. And as it can be seen it’s corresponding panel in 3C, it represents highest number of patients compared to number of patients in any other group. Which means that recurrence in 70% of large number of patients (almost 30?) is not negligible, that in terms of total number of patients with recurrence in low/high panel (12?). In simple words 70% of 30 patients, and 85 of 12 patients. describe/discuss this 70% recurrence group in discussion section. 

8.      Finally, while authors introduce, cancer stem cells (CSCs) at line 78; I suggest authors to refer here the following article (PMID: 29167804), which describes CSC clonal tumor initiation models (hierarchical model and stochastic model) and CSC markers.

Author Response

Reviewer #3

In this article Hou et al., report that low infiltration of CD8+ T cells in relation to high expression of PD-L1 are associated tumor recurrence in patients with pancreatic cancer (PC). Authors also show that low infiltration of T cells/high expression of PD-L1 is associated with levels of CD44+/CD133+ cancer stem cells and conclude that this correlation of infiltration/high expression in association with CD44+/CD133+ cancer stem cells could help predict an unfavorable prognosis in patients with PC.

The current study has a scope in a view that the study is performed on patient samples and expression correlates with tumor recurrence in patients and survival rate. The experimental design is technically sound and conclusions are supported with experimental evidence. However, there are some points that authors need to revise.

1)     The part of title which says ‘’….expression are associated with cancer stem cells and…’’. Indicate what capacity of cancer stem cells?. i.e. use an appropriate biological term with cancer stem cells e.g. any of these: cancer stem cell initiation, or cancer stem cell expression, or cancer stem cell levels, or cancer stem cell expansion.

Ans: Thank you for your comment and suggestion. The title “Low CD8+ T cell infiltration and high PD-L1 expression are associated with CD44+/CD133+ cancer stem cells and predict an unfavorable prognosis in pancreatic cancerhas been changed to “Low CD8+ T cell infiltration and high PD-L1 expression are associated with level of CD44+/CD133+ cancer stem cells and predict an unfavorable prognosis in pancreatic cancer”. Please check it.

2)     Same at page 4, line 135: CD44+/CD133+ CSCs, and Line 145; CD44+/CD133+ CSCs. Apply appropriate term with CSCs throughout the manuscript, including figure legends.

Ans: Thank you for your comment. The term “CD44+/CD133+ CSCs” has been changed to “CSCs”. Please check it in page 4, lines 143 and 152, and page 17, line 564 (figure legends).

3)     Line 31 of abstract: ‘’Moreover, CD8+ T cells infiltration could reduce unfavorable prognostic effect of high co-expression of PD-L1 and CD44/CD133’’. Perhaps better to mention high infiltration (high infiltration of CD8+ T cells).

Ans: Thank you for your suggestion. The sentence “Moreover, CD8+ T cells infiltration could reduce unfavorable prognostic effect of high co-expression of PD-L1 and CD44/CD133” has been changed to “Moreover, high infiltration of CD8+ T cells could reduce unfavorable prognostic effect of high co-expression of PD-L1 and CD44/CD133”. Please check it in page 1, line 31.

4)     Page 5, line 153; ‘’percentage of recurrent PC in patients with tumors exhibiting low CD8+ T cells infiltration….’’. Authors may wish to rephrase it as ‘’percentage of recurrent PC in patients group exhibiting low……’’.

Ans: Thank you for your suggestion. The sentence “percentage of recurrent PC in patients with tumors exhibiting low CD8+ T cells infiltration…” has been replaced with “the percentage of recurrent PC in the patient group exhibiting low CD8+ T cells infiltration/high PD-L1 expression was the highest (85%) than that in those with high CD8+ T cells infiltration/high PD-L1 expression (70%)…”. Please check it in page 5, lines 160-161.

5)     Figure 2 B. were the number of patients same in both low PD-L1 expression group and low PD-L1 expression group for evaluating the percentage of recurrence. In legends, mention the total number of patients included in each group.

Ans: Thank you for your suggestion. We have mentioned the number of patients in each group as recommended. Please check it in page 15, line 549 (figure legends).

6)     In Figure 4C (CD8 T cell infiltration/PD-L1 expression; low/high panel for CD 44/CD133 CSCs) and Figure 4D (CD8 T cell infiltration/PD-L1 expression; low/high panel for recurrence): Does this low/high panel in figure 4D correspond to same NUMBER of patients in 4C?. Perhaps mentioning the number of patients in 4D will be a suitable presentation, (however Figure 4C is self/explanatory by dot plots). This is important to make reader understandable whether this panel low/high in 4D corresponds to same patients and same NUMBER of patients when it comes to represent percentage.

Ans: Thank you for your comment and suggestion. The same patients and same case numbers in each group for Figure 4C is corresponding to Figure 4D. Please check in page 17, lines 568-575 (figure legends): (C, D) Classification of tumors based on CD8+ T cells infiltration and PD-L1 expression into 4 groups including low CD8+ T cells infiltration/low PD-L1 expression (n=30), high CD8+ T cells infiltration/low PD-L1 expression (n=13), low CD8+ T cells infiltration/high PD-L1 expression (n=13), and high CD8+ T cells infiltration/high PD-L1 expression (n=30). (C) Dot plots showing the percentage of CD44+/CD133+ CSCs in each group. All values are the mean ± s.e.m. NS, not significant; **, P<0.01 compared with low CD8+ T cells infiltration/low PD-L1 expression group, as determined using one-way ANOVA. (D) Relative distribution analysis of PC patients’ recurrence status within each group. Absent, patients without recurrent PC; present, patients with recurrent PC. P values calculated using Chi-square test. CSCs, cancer stem cells.

7)     Figure 4D high/high panel shows 70% of the cases exhibit recurrence. And as it can be seen it’s corresponding panel in 3C, it represents highest number of patients compared to number of patients in any other group. Which means that recurrence in 70% of large number of patients (almost 30?) is not negligible, that in terms of total number of patients with recurrence in low/high panel (12?). In simple words 70% of 30 patients, and 85 of 12 patients. describe/ discuss this 70% recurrence group in discussion section.

Ans: Thank you for your comment. In this study, we observed that the percentage of recurrent PC was the highest in the low CD8+ T cells infiltration/high PD-L1 expression group (85%, n = 13) followed by the high CD8+ T cells infiltration/high PD-L1 expression group (70%, n = 30), the high CD8+ T cells infiltration/low PD-L1 expression group (61%, n = 13), and the low CD8+ T cells infiltration/low PD-L1 expression group (60%, n = 30). Although there was a little difference among these groups and the results may be influenced by the small sample sizes, we found that PC patients with low CD8+ T cells infiltration and high PD-L1 expression experienced the worst OS and DFS, which is consistent with a previous study using a PC database from The Cancer Genome Atlas [1]. Indeed, the combination of CD8+ T cells infiltration and high PD-L1 expression stratifies patient survival. Please check in page 5, lines 205-213: changed as recommended.

8)     Finally, while authors introduce, cancer stem cells (CSCs) at line 78; I suggest authors to refer here the following article (PMID: 29167804), which describes CSC clonal tumor initiation models (hierarchical model and stochastic model) and CSC markers.

Ans: Thank you for your comment and suggestion. In page 2, lines 79-83, “The existence of a CSC population establishes a functional hierarchy within a tumor tissue and encompasses both the self-renewal and differentiation hallmarks of stem cells [2]. Pancreatic CSCs were identified based on a number of cell surface markers including CD24, CD44, CD133, ESA, c-Met, CXCR4, and ALDH1 [3]” is added as recommended.

References

1.        Zheng, W.; Skowron, K.B.; Namm, J.P.; Burnette, B.; Fernandez, C.; Arina, A.; Liang, H.; Spiotto, M.T.; Posner, M.C.; Fu, Y.X., et al. Combination of radiotherapy and vaccination overcomes checkpoint blockade resistance. Oncotarget 2016, 7, 43039-43051, doi:10.18632/oncotarget.9915.

2.       Dalerba, P.; Cho, R.W.; Clarke, M.F. Cancer stem cells: models and concepts. Annu Rev Med 2007, 58, 267-284, doi:10.1146/annurev.med.58.062105.204854.

3.        Abel, E.V.; Simeone, D.M. Biology and clinical applications of pancreatic cancer stem cells. Gastroenterology 2013, 144, 1241-1248, doi:10.1053/j.gastro.2013.01.072.

Round 2

Reviewer 1 Report

.

Reviewer 2 Report

Revised manuscript is now acceptable for the publication of "Cancers".

Reviewer 3 Report

Authors have revised their manuscript in the light of reviewrs comments, and have improved the content significantly. 

I have no further comments